# EXPRESSIVENESS OF NEURAL NETWORKS HAVING WIDTH EQUAL OR BELOW THE INPUT DIMENSION

## ABSTRACT

The understanding about the minimum width of deep neural networks needed to ensure universal approximation for different activation functions has progressively been extended (Park et al., 2021). In particular, with respect to approximation on general compact sets in the input space, a network width less than or equal to the input dimension excludes universal approximation. In this work, we focus on network functions of width less than or equal to the latter critical bound. We prove a maximum principle from which we conclude that for all continuous and monotonic activation functions, universal approximation of arbitrary continuous functions is impossible on sets that coincide with the boundary of an open set plus an inner point. Conversely, we prove that in this regime, the exact fit of partially constant functions on disjoint compact sets is still possible for ReLU network functions under some conditions on the mutual location of these components.

## 1 INTRODUCTION

In the course of the increasing popularity of deep neural networks for applications in technical, ecological and many other fields, there has been huge progress in the research on understanding the mathematical properties of the mathematical mapping implemented by a deep neural network. The approximation properties, or expressiveness, of neural network functions have attracted intense interest. The central result in this field is the classic *universal approximation theorem*, which states that any continuous function can be approximated with arbitrary accuracy (in terms of uniform approximation or $L_p$ norms) by neural network functions that have only one hidden layer for nearly every activation function (Cybenko, 1989; Hornik, 1991). In recent years, these kind of results have become more precise, for instance, in terms of estimates on the order of magnitude of parameters needed to achieve an approximation of a certain accuracy (Petersen & Voigtlaender, 2018; Yarotsky, 2017; 2018), or by investigating the number off piece-wise linear regions (Montufar et al., 2014; Serra et al., 2018; Hanin & Rolnick, 2019); we refer to (DeVore et al., 2020) for an overview on recent developments. Since empirical observations showed that depth has a significant impact on the performance of neural networks, a lot of research has been dedicated to the effect of depth on the expressive power of neural networks (Telgarsky, 2016; Mhaskar & Poggio, 2016; Montufar et al., 2014; Raghu et al., 2017; Rolnick & Tegmark, 2018; Lin et al., 2017; Cohen et al., 2016). Besides this, the role of width in expressiveness of network functions has been investigated (Lu et al., 2017; Hanin & Sellke, 2017; Hanin, 2019; Johnson, 2018; Kidger & Lyons, 2020; Park et al., 2021). An important result states that a width larger than the input dimension is needed to allow universal approximation on arbitrary compact sets. It has also been shown that the capability to learn disconnected or bounded decision regions depends on whether the width is larger than the input dimension (Nguyen et al., 2018; Beise et al., 2021).

In a further line of research, the capabilities of neural networks to memorize finite sample sets is investigated (Yun et al., 2019; Vershynin, 2020; Bubeck et al., 2020; Park et al., 2020), which provides mathematical foundations to the surprising findings that neural networks can interpolate the training data (Zhang et al., 2017) in practise.

In this work, we investigate what kind of subsets $M \subset \mathbb{R}^n$ admit universal approximation by neural network functions with a maximum layer width less than or equal to the input dimension, i.e. network functions that violate the necessary condition for universal approximation given in Hanin (2019) and Park et al. (2021) for arbitrary compact sets.

Let us introduce the following notation: For a set $D \subset \mathbb{R}^n$ we denote by $D^\circ$ the set of interior points, by $\overline{D}$ its closure and by $\partial D = \overline{D} \setminus D^\circ$ the boundary of $D$. By $\|\cdot\|$ we mean the Euclidean norm. For a linear subspace $U \subset \mathbb{R}^n$, we denote by $\dim(U)$ its dimension and by $P_U : \mathbb{R}^n \to U$ the orthogonal linear projection on $U$. We denote by $e_j$ the unit vector of the $j$-th coordinate axis in $\mathbb{R}^n$. For some $v \in \mathbb{R}^n$ and $j \in \{1, ..., n\}$, we write $v^{(j)}$ for the $j-$th component of $v$.

For some depth $L \in \mathbb{N}$, we consider neural network functions of some , $F : \mathbb{R}^{n_0} \to \mathbb{R}^{n_L}$ where $n_0$ is called the input dimension and $n_L$ the output dimension. Our network functions have the following form

$$F := W_L(A_{L-1} \circ ... \circ A_1) + b_L \qquad (1)$$

where $A_j(x) = \sigma(W_j x + b_j)$ with $W_j \in \mathbb{R}^{n_j \times n_{j-1}}$ (weights), $b_j \in \mathbb{R}^{n_j}$ (bias), where $j = 1, ..., L$, and $\sigma : \mathbb{R} \to \mathbb{R}$ the activation function. The application of $\sigma$ and the inverse image $\sigma^{-1}$ are understood to be applied element wise to vectors or subsets of $\mathbb{R}^n$. The widely used activation function *rectified linear unit*, shortly ReLU, is defined by $t \mapsto \max\{t, 0\}$. As a shorthand notation of the function implemented by the first $k$ layers of a network function, we set

$$F_k = W_k (A_{k-1} \circ ... \circ A_1) + b_k, \qquad (2)$$

for $k \in \{1, ..., L-1\}$. We call $n_j$ the *width* of layer $j = 1, ..., L$. The width of the network is defined as $\omega(F) = \max\{n_j : j = 1, ..., L\}$ and $L$ is called the depth of the network. For $m, L \in \mathbb{N}$ and an activation function $\sigma$, $\mathcal{NN}_\sigma^k(m, L)$ is the set of network functions $F : \mathbb{R}^{n_0} \to \mathbb{R}^k$ of the form (1) with activation $\sigma$ and of maximum width $m$, i.e. $\omega(F) \leq m$, and depth $L$, i.e. $L$ layers including the final linear layer. In case that $\mathbb{R}^k = \mathbb{R}$ we omit the latter and write $\mathcal{NN}_\sigma(m, L)$ instead, and in case the the depth is not specified, meaning that arbitrary depth is allowed, we write $\mathcal{NN}_\sigma^k(m)$, $\mathcal{NN}_\sigma(m)$, respectively.

We say that a compact subset $M \subset \mathbb{R}^{n_0}$ admits universal uniform approximation of functions in some function class of mappings from $M$ to $\mathbb{R}$ by network function of $\mathcal{NN}_\sigma(m, L)$ or $\mathcal{NN}_\sigma(m)$, if for every $f : M \to \mathbb{R}$ in that class and every $\varepsilon > 0$, there exists some $F \in \mathcal{NN}_\sigma(m, L)$, respectively $F \in \mathcal{NN}_\sigma(m)$, such that

$$\max_{x \in M} |f(x) - F(x)| < \varepsilon. \qquad (3)$$

## 2 RELATED WORK

Approximation properties of width bounded neural networks have been studied in several papers. A common goal in these lines of research is to provide upper and lower bounds on the minimum width of the network, i.e. the minimum number of neurons per layers, here denoted by $\omega_{\min}$, needed for universal approximation in $C(M, \mathbb{R}^{n_L})$, i.e. the space of continuous functions from a compact set $M$ to $\mathbb{R}^{n_L}$ endowed with the norm of uniform convergence, and $L_p(M, \mathbb{R}^{n_L})$ with some set $M \subset \mathbb{R}^{n_0}$, i.e. the space of functions $f = (f_1, ..., f_{n_L})$ from $M$ to $\mathbb{R}^{n_L}$ such that every $|f_j|^p$ is integrable over $M$, endowed with the usual $L_p$ norm. Since in this work we are interested in universal approximation in $C(M, \mathbb{R})$, we refer to Lu et al. (2017), Kidger & Lyons (2020) and Park et al. (2021) for the case of $L_p$ approximation. For the case of $C(M, \mathbb{R}^{n_L})$, it was proven in Hanin & Sellke (2017) that

$$n_0 + 1 \leq \omega_{\min} \leq n_0 + n_L, \text{ for ReLU activation}, \qquad (4)$$

which has been tightened in Park et al. (2021) to

$$\omega_{\min} \geq \max\{n_0 + 1, n_L\} \text{ for ReLU and STEP activation}, \qquad (5)$$

where STEP refers to the threshold activation that maps to 1 on $\{x \geq 0\}$ and 0 otherwise. It has been shown at the same time in Johnson (2018) and Beise et al. (2021) that

$$\omega_{\min} \geq n_0 + 1 \text{ for ReLU + injective continuous activation}. \qquad (6)$$

The activation functions allowed in Johnson (2018) are slightly more general, as it is only required that they admit arbitrary accurate uniform approximation by injective continuous functions on arbitrary compact subset of $\mathbb{R}$. In Kidger & Lyons (2020) the above upper bounds are extended to further classes of activation functions. For these results and a general overview on recent developments in this field, we refer to Park et al. (2021). As we also formulate results for finite sets, our work partially exhibits relations to the research on finite sample memorization (Yun et al., 2019; Vershynin, 2020; Bubeck et al., 2020; Park et al., 2020).

In this work, we only consider width less than or equal to $n_0$ and investigate uniform approximation on certain compacts subsets of the input space. According to the above result (6), we cannot expect universal approximation for all compact $M \subset \mathbb{R}^{n_0}$. However, a common assumption in applications is that approximation is only needed on a certain subset. We derive some topological conditions on $M$ that allow or exclude a kind of universal approximation under these circumstances. Although narrow neural networks, as they are considered in this work, are not common in practical applications, many networks exhibit a decaying layer width at the later layers. From this perspective, our results give theoretical insights on the kind feature extraction earlier layers need to implement in order to allow the later layers to solve a given machine learning task. Our main contributions are as follows:

1. A maximum principle is given for $F \in \mathcal{NN}_\sigma(n_0)$ with $\sigma$ continuous and monotonic. This allows to conclude that the lower bound $\omega_{\min} \geq n_0 + 1$ in (6) is sharp for a wide range of subsets $M$, e.g. when $M = \partial D \cup \{c\}$ for some $D$ with non-empty interior and $c \in D^\circ$.

2. We show that for the case of two disjoint compact sets, the existence of a cone-like sector that contains one of these sets and does not intersect with the other one, is sufficient to allow exact fit of functions that take constant values on each of these sets by network functions from $\mathcal{NN}_{\mathrm{ReLU}}(n_0, 4)$. A weaker result is concluded for the case of multiple pairwise compact components.

## 3    MAXIMUM PRINCIPLE

Let us recall that $\mathcal{NN}_\sigma(n_0)$ designates the set of network functions with activation function $\sigma$, having maximum width $n_0$ and arbitrary depth. In this section, we will prove a maximum principle for network functions of $\mathcal{NN}_\sigma(n_0)$ for a wide class of activation functions. This principle can be viewed as a root cause why universal approximation with functions from $\mathcal{NN}_\sigma(n_0)$ on arbitrary compact sets is impossible as shown in Johnson (2018) and Beise et al. (2021), see (6). This also immediately leads to a topological condition on subsets of $\mathbb{R}^{n_0}$ that do not admit universal approximation by functions in $\mathcal{NN}_\sigma(n_0)$.

**Theorem 1** (Maximum Principle). *Let $M$ be some compact subset of $\mathbb{R}^{n_0}$ and $\sigma$ a continuous, monotonic activation function. Then every $F \in \mathcal{NN}_\sigma(n_0)$ takes its maximum value at the boundary $\partial M$.*

Note that, considering $-F$ instead of $F$, the latter result also implies that the minimum value is taken on $\partial M$. This implies that universal uniform approximation of continuous functions is impossible on compact sets with non-empty interior. Even more, it can be concluded that:

**Corollary 1.** *Let $M$ be some compact subset of $\mathbb{R}^{n_0}$ such that $\partial D \subset M$ and $M \cap D^\circ \neq \emptyset$ for some non-empty open set $D \subset \mathbb{R}^n$. Then $M$ does not admit universal uniform approximation of continuous function by network functions from $\mathcal{NN}_\sigma(n_0)$.*

The latter results naturally lead to the following *question*: Is it possible to uniformly approximate arbitrary continuous functions on the unit sphere (in $\mathbb{R}^{n_0}$) with arbitrary accuracy by network functions as they are considered in Theorem 1? Some positive approximation results are given in the next section, but a gap to an answer to the question remains.

It is not clear to us whether the conditions of Theorem 1 can be weakened in a way that it also applies to network function having width larger than the input dimensions under certain circumstances. The following might be a natural question in this context: Let $F : \mathbb{R}^{n_0} \to \mathbb{R}$ be a neural network function of arbitrary width but with weight matrices having rank less than or equal to $n_0$. Does a maximum principle similar to Theorem 1 apply in this case? The following example shows that the maximum principle in Theorem 1 does not hold in this case.

**Example 1.** *Let $M = [-1, 1]$ and let $W_1 \in \mathbb{R}^{2 \times 1}$ and $b_1 \in \mathbb{R}^2$ implement the following linear affine mapping*

$$W_1 x + b_1 = \begin{pmatrix} x + 1 \\ x \end{pmatrix}.$$

*Then $M_1 = \mathrm{ReLU}(W_1 M + b_1)$ is given by $M_1 = M_{1,1} \cup M_{1,2}$ where*

$$M_{1,1} := \{(x + 1, 0)^T : x \in M, \text{and } x < 0\}$$

$$M_{1,2} := \{(x+1, x)^T : x^T \in M, \text{and } x \geq 0\}.$$

*Now, let $v = (1, 1/2)^T \in \mathbb{R}^2$, $W_2 \in \mathbb{R}^{2 \times 2}$ and $b_2 \in \mathbb{R}^2$ such that $x \mapsto W_2 x + b_2$ implements the orthogonal projection onto the hypersurface $V := \{x \in \mathbb{R}^2 : x^T v = (1, 0)v\}$. That is, the hypersurface orthogonal to $v$ which intersects $(1, 0)^T$. Then both, $M_{1,1}$ and $M_{1,2}$ are mapped to the line $M_2 := \{y \in \mathbb{R}^2 : y = (1, 0) + \lambda(-1/2, 1), 0 \leq \lambda \leq 1/2\}$ and since those vectors are contained in the first quadrant we have*

$$M_2 = \mathrm{ReLU}(W_2(\mathrm{ReLU}(W_1 M + b_1) + b_1).$$

*It can further easily be verified that, under the latter two layer network function, $0$, which is an inner point of $M$, is the only vector that is mapped to the extreme point $(1, 0)$ of $M_2$. Now it is easy to find a weight $W_3 \in \mathbb{R}^2$ such that the linear map $x \mapsto W_3 x$ as a mapping from $M_2$ to $\mathbb{R}$ takes its minimum or maximum value at $(1, 0)$ only.*

*By concatenation of additional components to the input dimension such that $M = [-1, 1] \times [0, 1]^d$, $d \in \mathbb{N}$, and adaptation of the weights $W_1, W_2$ and bias $b_1, b_2$ in a way that these additional components are mapped identically under the action of the two layer network function $x \mapsto \mathrm{ReLU}(W_2(\mathrm{ReLU}(W_1 x + b_1) + b_1)$, the above example can be extended to higher dimensions.*

The following proposition provides the main observation for the proof of Theorem 1.

**Proposition 1.** *Let $M$ be a compact subset of $\mathbb{R}^{n_0}$, $\sigma$ a continuous, monotonic activation function and $F \in \mathcal{NN}^{n_0}_\sigma(n_0)$ such that all weight matrices $W_j$, $j = 1, ..., L$, are square and have full rank. If for some $x \in M^\circ$ the image $F(x)$ is a boundary point of $F(M)$, i.e. $F(x) \in \partial F(M)$, then there is an $\tilde{x} \in \partial M$ such that $F(\tilde{x}) = F(x)$.*

*Proof.* In case that $\sigma$ is injective, the whole network function $F$ is injective since the weight matrices are assumed to be square and of full rank. Hence, the invariance of the domain theorem gives that no $x \in M^\circ$ can be mapped to $\partial F(M)$. The remaining case is that $\sigma$ is partially constant. Then, given an $x \in M^\circ$ with $F(x) \in \partial F(M)$, let $\tilde{k}$ be the smallest $k \in \{1, ..., L-1\}$ such that $\sigma(F_{\tilde{k}}(x)) \in \partial\sigma(F_{\tilde{k}}(M))$. Notice that such a $\tilde{k}$ must exist since, according to the invariance of the domain theorem, the last injective, linear affine mapping $x \mapsto W_L x$ cannot map inner points to boundary points. Then, necessarily for at least one component of $F_{\tilde{k}}(x)$, a partially constant part of $\sigma$ is active. More explicitly, say $(F_{\tilde{k}}(x))^{(j)} \in [a_j, b_j] \subset \mathbb{R}$ with $\sigma(t) = c_j$, where $c_j$ is the constant value taken for all $t \in [a_j, b_j]$, for some indices $j \in I \subset \{1, ..., n_0\}$. In case that one of those intervals can be enlarged arbitrarily to either $[a_j, \infty)$ or $(-\infty, b_j]$, which would be the case for ReLU activation, we can find a $t \in \mathbb{R}$ such that $x_{\tilde{k}} = F_{\tilde{k}}(x) + t e_j \in \partial F_{\tilde{k}}(M)$ and with $\sigma(x_{\tilde{k}}) = \sigma(F_{\tilde{k}}(x))$. In a next step we show that such an $x_{\tilde{k}}$ can also be found in cases where the length of the intervals $[a_j, b_j]$ are upper bounded. In those cases we may assume that the intervals $[a_j, b_j]$ are maximum in the sense that at their left and right end $\sigma$ is not constant any more, and $I$ is maximum in the sense that for every other index, the activation function $\sigma$ is injective in a neighbourhood. Indeed, if such components would not exist, the component wise application of $\sigma$ would be injective in a small neighbourhood of $F_{\tilde{k}}(x)$ and the invariance of the domain theorem would exclude that $\sigma(F_{\tilde{k}}(x)) \in \partial\sigma(F_{\tilde{k}}(M))$. By the selection of $\tilde{k}$, $F_{\tilde{k}}(x)$ is still an inner point of $F_{\tilde{k}}(M)$. The fact that the inner point $F_{\tilde{k}}(x) \in (F_{\tilde{k}}(M))^\circ$ is mapped to a boundary point by the element wise application of $\sigma$ implies that there is some non-empty $\tilde{I} \subset I$ with corresponding $t_j \in \mathbb{R}$ such that

$$t_j \in [-(F_{\tilde{k}}(x))^{(j)} + a_j, -(F_{\tilde{k}}(x))^{(j)} + b_j], \ j \in \tilde{I} \tag{7}$$

$$x_{\tilde{k}} := F_{\tilde{k}}(x) + \sum_{j \in \tilde{I}} t_j e_j \in \partial F_{\tilde{k}}(M). \tag{8}$$

Indeed, otherwise,

$$F_{\tilde{k}}(x) + \sum_{j=1}^{n_0} t_j e_j \in (F_{\tilde{k}}(M))^\circ$$

where for some small $\varepsilon > 0$ and all $j \in I$, $t_j$ may be chosen arbitrarily in $(-(F_{\tilde{k}}(x))^{(j)} + a_j - \varepsilon, -(F_{\tilde{k}}(x))^{(j)} + b_j + \varepsilon)$ and $\varepsilon > |t_j| > 0$ for $j \notin I$. By the definition of $I$ and the corresponding

intervals $[a_j, b_j]$, this would imply that $\sigma\left(F_{\tilde{k}}(x)\right)$ is an inner point of $\sigma\left(F_{\tilde{k}}(M)\right)$. Now, (8) implies that there is a $\tilde{x} \in M$ such that

$$F_{\tilde{k}}(\tilde{x}) = F_{\tilde{k}}(x) + \sum_{j=1}^{n_0} t_j e_j \in \partial F_{\tilde{k}}(M),$$

with $t_j$ as in (7). Considering that $W_{\tilde{k}}$ is square and has full rank, the inverse image of $F_{\tilde{k}}(\tilde{x})$ under $x \mapsto W_{\tilde{k}}x + b_{\tilde{k}}$ is a boundary point of $\sigma(F_{\tilde{k}-1}(M))$ in case that $\tilde{k} > 1$ and a boundary point of $M$ in case when $\tilde{k} = 1$. In the last case we are done. In the first case, the same reasoning can be applied to $\tilde{x}$ instead of $x$, and the iterative application finally leads to a boundary point of $M$ that is mapped to $\sigma\left(F_{\tilde{k}}(\tilde{x})\right) = \sigma\left(F_{\tilde{k}}(x)\right)$ and hence follows the same trajectory when passed through the last layers from $\tilde{k}$ to $L$. $\qquad\square$

*Proof.* (Theorem 1) We may assume that $M^\circ$ is non-empty and that every weight matrix $W_j$, $j = 1, ..., L - 1$, is square and of full rank. Otherwise the result is trivial, because $F(M) = \partial F(M)$. Hence the assumption of Proposition 1 holds for $\tilde{F} = A_{L-1} \circ ... \circ A_1$. Since the linear mapping $x \mapsto W_L x$ from $\mathbb{R}^{n_{L-1}}$ to $\mathbb{R}$ takes its maximum and minimum values on $\partial \tilde{F}(M)$, the latter proposition also implies that $F$ takes its maximum value on $\partial M$. $\qquad\square$

## 4 APPROXIMATION PROPERTIES ON SUBSETS

In this section we give some approximation results that show that, despite the limitations induced by Theorem 1, the network functions considered in this work still allow some weaker kind of universal approximation on subsets. The first result in this section follows from existing results on approximation by network functions. The first point can be deduced from Hanin & Sellke (2017, Theorem 1). The second point can be derived from Hanin & Sellke (2017, Proposition 2) or from Hardt & Ma (2017, proof of Theorem 3.2.).

**Proposition 2.**

1. *Let $M$ be some compact subset of $\mathbb{R}^{n_0}$. If there exists a subspace $U \subset \mathbb{R}^{n_0}$ with $\dim(U) < n_0$ such that $P_U$ is injective as a mapping from $M$ to $U$, then $M$ admits universal uniform approximation of continuous function by network functions in $\mathcal{NN}_{\mathrm{ReLU}}(\dim(U) + 1)$.*

2. *Let $M = \{x_1, ..., x_m\} \subset \mathbb{R}^{n_0}$ be a finite set and $f : M \to \mathbb{R}$ some mapping. Then there exists a network function $F \in \mathcal{NN}_{\mathrm{ReLU}}(2)$ such that $F(x) = f(x)$ for all $x \in M$.*

The proof is almost straightforward, given the results from the aforementioned works. A proof can be found in the appendix.

We next extend the second statement of the latter proposition to the case of two disjoint $n_0$-dimensional compact sets. Theorem 1 shows that arbitrary uniform approximation is not admitted on such domains. However, exact fit is possible for functions that are constant on each of these sets under certain geometrical properties on their mutual location. The approximation of such functions arises naturally in classification tasks where each point in a particular disjoint compact set belongs to the same class, i.e. these sets constitute different disjoint clusters of the data.

**Theorem 2.** *Let $M = K_1 \cup K_2$, with disjoint compact sets $K_1, K_2 \subset \mathbb{R}^{n_0}$ such that for some $c \in \mathbb{R}^{n_0}$ and some linearly independent $v_1, ..., v_{n_0} \in \mathbb{R}^{n_0}$ the set $K_1$ is contained in the sector*

$$S := \{x \in \mathbb{R}^{n_0} : x = c + \sum_{j=1}^{n_0} \lambda_j v_j, \ \lambda_j > 0\}$$

*and $K_2 \subset \mathbb{R}^{n_0} \setminus \overline{S}$. Then for every function $f : M \to \mathbb{R}$ that takes constant values on $K_1$ and $K_2$, respectively, there exists a network function $F \in \mathcal{NN}_{\mathrm{ReLU}}(n_0, 4)$ such that*

$$f(x) = F(x)$$

*for every $x \in M$.*

The conditions of Theorem 2 are verified and inspected by numerical experiments in Section 6. However, to us it is unclear whether the assumptions on the mutual location of the compact components in Theorem 2 can be significantly weakened. Increasing the depth will certainly allow more complex configurations. However, from Theorem 1 is clear that $K_2$ cannot completely enclose $K_1$ when the maximum width is upper bounded by $n_0$, no matter how many layers the network contains. We think that closing the described gap could entail important insights for the theory of neural networks.

For the proof of the latter result we need the following.

**Proposition 3.** *Let $K$, $M$ be two compact sets in $\mathbb{R}^{n_0}$ that are strictly separable by a linear hyper surface. Then there exists an $F \in \mathcal{NN}_{\mathrm{ReLU}}^{n_0}(n_0, 2)$ such that $F(K)$ is a single vector in $\mathbb{R}^{n_0} \setminus M$ and $F(x) = x$ for all $x \in M$. Moreover, $F$ can be arranged in a way that given $\varepsilon > 0$*

$$\min_{x \in K} \|F(K) - x\| < \varepsilon.$$

As a consequence of Proposition 3 and Proposition 2, we obtain the following for the case of more than two distinct sets under considerably more restrictive condition as in Theorem 2.

**Corollary 2.** *Let $M = \bigcup_{j=1}^d K_j$ with pairwise disjoint compact sets $K_j \subset \mathbb{R}^{n_0}$, $j = 1, ..., d$ such that for every $K_j$ there exist $v_j \in \mathbb{R}^{n_0}$ and $q_j \in \mathbb{R}$ such that*

$$v_j^T x > q_j \text{ for all } x \in K_j, \text{ and } v_j^T x < q_j \text{ for all } x \in M \setminus K_j,$$

*i.e. there are linear hypersurfaces separating each $K_j$ from the remaining $K_l$, $l \neq j$. Then for every function $f : M \to \mathbb{R}$ that takes constant values on every $K_j$, $j = 1, ..., d$, there exists a network function $F \in \mathcal{NN}_{\mathrm{ReLU}}(n_0)$ with*

$$f(x) = F(x)$$

*for every $x \in M$.*

*Proof.* (Corollary 2) By means of Proposition 3, we find an $F_1 \in \mathcal{NN}_{\mathrm{ReLU}}^{n_0}(n_0)$ such that $F_1(x) = x$ for $x \in \bigcup_{j=2}^d K_j$ and $a_1 := F_1(K_1)$ is a single vector in $\mathbb{R}^{n_0} \setminus \bigcup_{j=2}^d K_j$. Proposition 3 also allows us to choose $F_1$ in a way that $a_1$ is sufficiently close to $K_1$ that the conditions of Corollary 2 still apply to $\bigcup_{j=2}^d K_j \cup \{a_1\}$. We can iteratively apply Proposition 3 to find $F_1, ..., F_d \in \mathcal{NN}_{\mathrm{ReLU}}^{n_0}(n_0)$ such that

$$F_d \circ ... \circ F_1(K_j) = a_j, \; j = 1, ..., d$$

for pairwise distinct $a_1, ..., a_d \in \mathbb{R}^{n_0}$. Now, for a given $f : M \to \mathbb{R}$ with $f(x) = y_j$ for all $x \in K_j$, $j = 1, ..., d$, Proposition 2 yields an $\tilde{F} \in \mathcal{NN}_{\mathrm{ReLU}}(2)$ with $\tilde{F}(a_j) = y_j$ for all $j = 1, ..., d$. The desired network function is then obtained by $F := \tilde{F} \circ F_d \circ ... \circ F_1$. $\square$

*Proof.* (Proposition 3) Let us remind that $e_1, ..., e_{n_0}$ are the standard basis vectors in $\mathbb{R}^{n_0}$. By assumption, there is a $v \in \mathbb{R}^{n_0}$, with $\|v\| = 1$ and a $q \in \mathbb{R}$ such that $v^T x > q$ for $x \in K$ and $v^T x < q$ for $x \in M$. For a given $\varepsilon > 0$ we may assume that the linear hyper surface defined by $v, q$, $H := \{x \in \mathbb{R}^{n_0} : v^T x = q\}$ is so near to $K$ that for some $a \in K$ that realizes the minimum distance of $K$ to $H$ we have

$$v^T a = q + \varepsilon/2. \tag{9}$$

Indeed, otherwise we can shift the hyper surface accordingly by increasing $q$. Then, by the fact that $\|v\| = 1$, (9) means that $a$ has a distance equal to $\varepsilon/2$ to $H$ and $\tilde{a} := a - \varepsilon/2\,v \in H$ is the unique vector in $H$ that realizes this distance. Let $V_1 \in \mathbb{R}^{n_0 \times n_0}$ such that $x \mapsto V_1 x$ implements a length preserving rotation that maps $v$ to $-e_1$. Then $x \mapsto V_1 x - V_1 \tilde{a}$ maps $H$ to $\mathrm{span}(e_2, ..., e_{n_0})$ and $K_1 := V_1 K - V_1 \tilde{a}$ is contained in the half space of vectors with negative first component and with $a_1 := V_1 a - V_1 \tilde{a} = -\varepsilon/2\,e_1$, and $M_1 := V_1 M - V_1 \tilde{a}$ is contained in the half space of vectors with positive first component. By compactness of $K_1$ and $M_1$, we can find $u_1, ..., u_{n_0} \in \mathrm{span}(e_2, ..., e_{n_0})$, such that for a $\delta > 0$, $-u_1 - \delta e_1, ..., -u_{n_0} - \delta e_1$ are linearly independent and such that with $\delta$ sufficiently small

$$K_1 \subset S^- := \{x \in \mathbb{R}^{n_0} : x = \sum_{j=1}^{n_0} \lambda_j(-u_j - \delta e_1), \, \lambda_j > 0\}$$

$$M_1 \subset S^+ := \{x \in \mathbb{R}^{n_0} : x = \sum_{j=1}^{n_0} \lambda_j(u_j + \delta e_1), \, \lambda_j > 0\}.$$

Let $V_2 \in \mathbb{R}^{n_0 \times n_0}$ such that $x \mapsto V_2 x$ maps $-u_j - \delta e_1$ to $-e_j$ for $j = 1, ..., n_0$. Then, by linearity, every $u_j + \delta e_1$ is mapped to $e_j$ for $j = 1, ..., n_0$. Thus $S^-$ is mapped to the cone of vectors having negative components only, and $S^+$ is mapped to the cone of vectors having positive components only. Hence, the application of ReLU maps all of $V_2 K_1$ to 0 and maps $V_2 M_1$ identically. We can now apply the inverse of the linear affine mappings which will map $V_2 M_1$ back to $M$, i.e. for

$$F(x) := V_1^{-1} V_2^{-1} \mathrm{ReLU}(V_2(V_1 x - V_1 \tilde{a})) + \tilde{a}.$$

With this $F$ we have $F(M) = M$ and $F(K) = F(a) = \tilde{a}$. Note that by (9) and the definition of $\tilde{a}$

$$\min_{x \in K} \|x - F(a)\| = \varepsilon/2 < \varepsilon$$

which concludes the proof. $\qquad\qquad\square$

*Proof.* (Theorem 2) Let $V$ be the matrix that results from the concatenation of the columns $v_1, ..., v_{n_0}$ and set $W_1 := -V^{-1}$ and $b_1 := W_1(-c)$. Then the mapping $x \mapsto W_1 x$ maps the $v_j$ to $-e_j$, $j = 1, ..., n_0$ and $W_1 K_1 + b_1$ is a subset of

$$S^- := \{x \in \mathbb{R}^{n_0} : x = \sum_{j=1}^{n_0} \lambda_j(-e_j), \; \lambda_j > 0\}$$

and $W_1 K_2 + b_1 \subset \mathbb{R}^{n_0} \setminus \overline{S^-}$. Thus, the one layer network function $F_1(x) := \mathrm{ReLU}(W_1 x + b_1)$ maps all $x \in K_1$ to 0 and $x \in K_2$ are mapped to

$$\{x \in \mathbb{R}^{n_0} : x = \sum_{j=1}^{n_0} \lambda_j e_j, \; \lambda_j \geq 0\} \setminus \{0\}.$$

Hence, with a sufficiently small $q > 0$, $F_1(K_1)$ and $F_1(K_2)$ are strictly separated by the linear hyper surface

$$\{x \in \mathbb{R}^{n_0} : (1, 1, ...., 1)x = q\}.$$

The twofold application of Proposition 3 yields two network functions that can be concatenated to a three layer network function $F_2 \in \mathcal{NN}_{\mathrm{ReLU}}^{n_0}(n_0, 3)$ with $F_2(x) = W_4 \mathrm{ReLU}(W_3 \mathrm{ReLU}(W_2 x + b_2) + b_3) + b_4$ and such that $F_2(F_1(K_1))$ and $F_2(F_1(K_2))$ are two distinct vectors $u_1, u_2$ in $\mathbb{R}^{n_0}$, respectively. For a given $f : M \to \mathbb{R}$ that takes constant values on $K_1$ and $K_2$, say $a_1, a_2 \in \mathbb{R}$, respectively, we attach a linear affine layer $x \mapsto w^T x + b_5$, where $w \in \mathbb{R}^{n_0}$, $b_5 \in \mathbb{R}$ such that $w^T u_1 + b_5 = a_1$ and $w^T u_2 + b_5 = a_2$. This mapping is integrated in the final layer of $F_2 \circ F_1$ and finally gives

$$F(x) = w^T W_4 \mathrm{ReLU}(W_3 \mathrm{ReLU}(W_2 \mathrm{ReLU}(W_1 x + b_1) + b_2) + b_3) + w^T b_4 + b_5.$$

$\qquad\qquad\square$

We conclude this section with a comment on implications of our results on generalisation. In Zhang et al. (2017) the researchers made the observation that deep neural networks can interpolate randomly annotated data, which shows the enormous capabilities of large networks to memorize data. In Proposition 2, Theorem 2 and Corollary 3 we show that network functions of width less than or equal than to input dimension in many configurations also implement an exact fit of the training data. At the same time these network functions are restricted by the maximum principle given in Theorem 1, which may contradict the underlying distribution of the data at hand.

## 5    UNIQUENESS

A maximum principle similar to Theorem 1 also applies for harmonic and holomorphic functions, that is, the null space of the Laplace operator and the null space of $\partial_{\overline{z}} := 1/2(\partial_x + i\partial_y)$ ($x$ real part, $y$ imaginary part), respectively (Rudin, 2006). As a consequence, a uniqueness theorem applies which states that two harmonic, respectively holomorphic, functions coincide on the interior of a given set when they coincide on its boundary, respectively. In view of Theorem 1, it is hence natural to ask whether a similar result applies in the case of network functions in $\mathcal{NN}_\sigma(n_0)$. However, the proof for the case of harmonic and holomorphic functions relies on the fact that these functions form a vector space, which is not the case for the set $\mathcal{NN}_\sigma(n_0)$. The following examples show that a uniqueness theorem does not hold, in general, for the network functions with (strictly) monotonic activation functions considered in this work.

**Example 2.** *Let $\sigma_\alpha(x)$ be the leaky ReLU function defined by $\sigma_\alpha(x) = x$ for $x \geq 0$ and $\sigma_\alpha(x) = \alpha x$ for $x < 0$ with $\alpha > 0$. Then*

$$x \mapsto \sigma_\alpha \left( \frac{1}{(1+\alpha)} \sigma_\alpha(x) + \frac{\alpha}{(1+\alpha)} \right) \ and \ x \mapsto \sigma_\alpha(\frac{1}{2}\sigma_\alpha(x+1))$$

*both map $-1$ to $0$ and $1$ to $1$, but the first one maps $0$ to $\alpha/(1+\alpha)$, while the second one maps $0$ to $1/2$ and hence do not coincide unless $\alpha = 1$.*

**Example 3.** *A similar example can be obtained for analytic activation functions. Indeed, let $\sigma$ be the sigmoid function. For $x \mapsto \sigma(ax + b)$, one can arrange $a > 0$, and $b \in \mathbb{R}$, such that in one case only a concave and in a second case a convex excerpt of sigmoid is active for $x$ in [0,1], respectively, and such that both functions coincide at $0$ and $1$. As the first one is concave and the second one is convex on [0,1] and both are non-linear, they do not coincide on the interior of [0,1].*

A simple uniqueness result follows from the following observation.

**Proposition 4.** *Let $M$ be a compact subset of $\mathbb{R}^{n_0}$, $F \in \mathcal{NN}_{\mathrm{ReLU}}^{n_0}(n_0)$ and $B = F^{-1}\left( (F(M))^\circ \right)$ the inverse image of the inner points of $F(M)$. Then $F$ is linear affine and bijective as a mapping from $B$ to $(F(M))^\circ$.*

A proof of Proposition 4 is given in the appendix. As a consequence we have.

**Corollary 3.** *Let $F, G \in \mathcal{NN}_{\mathrm{ReLU}}^{n_0}(n_0)$ and $M \subset \mathbb{R}^{n_0}$ be some compact set. If $F$ and $G$ coincide on a set of vectors $v_1, ..., v_{n_0+1}$ in*

$$K := F^{-1}\left( (F(M))^\circ \right) \cap G^{-1}\left( (G(M))^\circ \right),$$

*such that $v_j - v_{n_0+1}$, $j = 1, ..., n_0$ are linearly independent, then $F(x) = G(x)$ for all $x \in K$.*

It should be noted that in the latter corollary, the fact that $F, G$ coincide on $K$ does not imply that $F$ and $G$ are the same network functions as functions on $\mathbb{R}^{n_0}$.

## 6 NUMERICAL EXPERIMENT

We provide numerical results to illustrate that neural networks learn transformations similar to the construction of our theoretical derivations in the proof of Theorem 2. We formulate a toy example consisting of several (6 or 8) pair-wise disjoint balls of the same class encasing a center ball of a different class. The border balls use a radius of $0.125$ and a subset of the centres $(0.25, 0.25)$, $(0.5, 0.25)$, $(0.75, 0.25)$, $(0.25, 0.5)$, $(0.75, 0.5)$, $(0.25, 0.75)$, $(0.5, 0.75)$ and $(0.75, 0.75)$ while the center ball uses a radius of $0.01$ and the center $(0.5, 0.5)$. We generated randomly and uniformly 2000 data points for each border ball, however, as the classes would hence be unbalanced, we generated as many data points for the center ball as there are in total for the border balls. We used a multilayer perceptron (MLP) consisting of 4 layers, all of width 2, and the $\mathrm{ReLU}$ activation function (as stated in the theorem). For illustration purposes, we used an input dimension of 2 such that the neural network functions are of the form $F \in \mathcal{NN}_{\mathrm{ReLU}}(2)$. We used a batch size of 16, the Adam optimizer with a learning rate of 0.001 and we trained each model for 500 epochs using the mean-squared error (MSE) as our cost function. The different datasets and results are reported in Fig. 1. We rate an experiment as successful when the universal uniform approximation condition (3) tends to zero. It can be observed in Fig. 1 (a) that, when the conditions of Theorem 2 are fulfilled, the neural network can approximate the function correctly and the different layers are learning transformations similar to the constructions in the proof. When the conditions are violated, as is shown in Fig. 1 (b), the universal uniform approximation condition does not tend to zero and the intermediate layer transformations need to violate the constructions in the proof. Additional results are reported in Fig. 2 in the appendix.

## 7 CONCLUSION

We identified a maximum principle that holds in the case of width less than or equal to the input dimension for all common activation functions, which can be interpreted as a root cause why universal approximation is not possible under the aforementioned conditions. On the positive side we proved

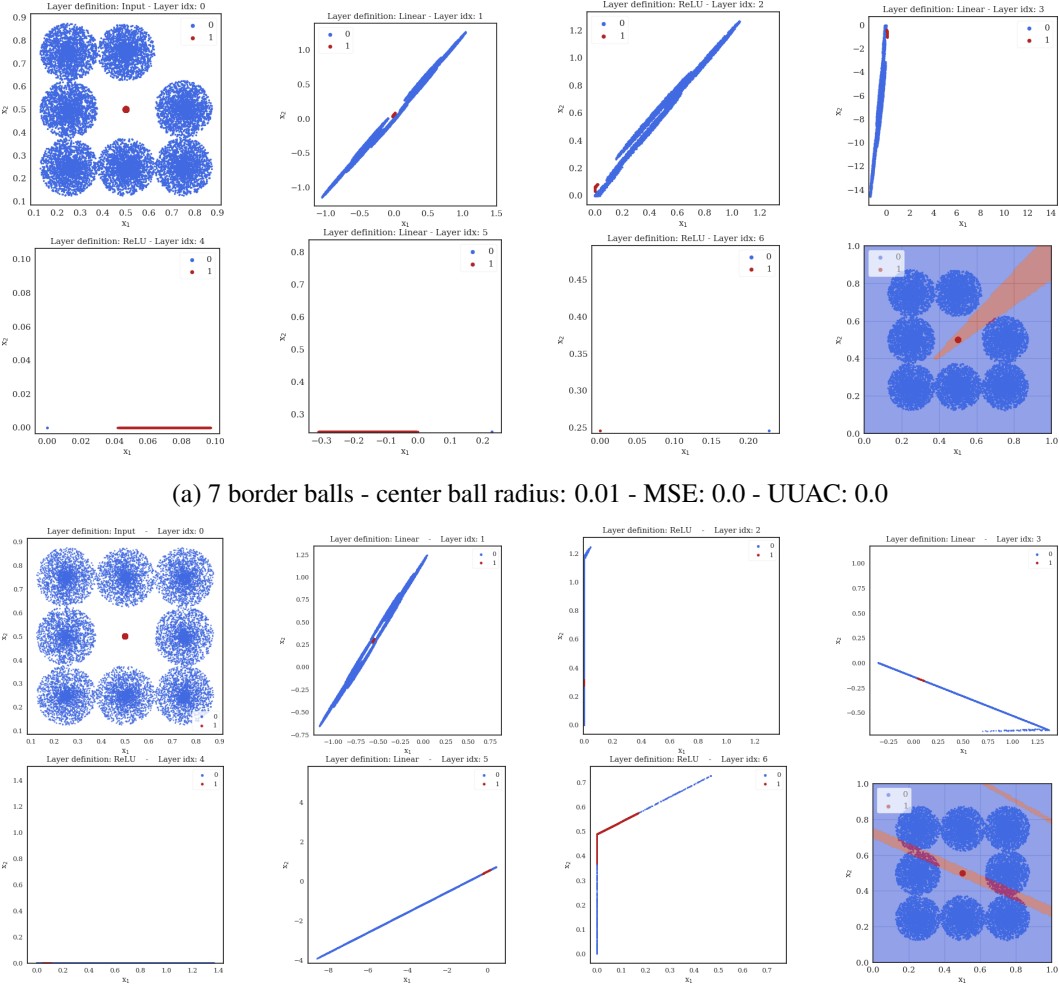

(a) 7 border balls - center ball radius: 0.01 - MSE: 0.0 - UUAC: 0.0

(b) 8 border balls - center ball radius: 0.01 - MSE: 0.01135 - UUAC: 1.08494

Figure 1: Datasets (a) and (b) together with the learned transformations by each activation function and layer. When the conditions of Theorem 2 are fulfilled (a), the transformations are similar to the construction in the proof. Otherwise (b), the construction in the proof is violated. We report the final decision regions, the mean-square error (MSE) and the universal uniform approximation condition (UUAC). Each subplot's title describes the function which was applied to the previous subplot's data.

that, although being not sufficient for universal approximation in the general sense, ReLU network functions of width less than or equal to the input dimension are sufficient to implement functions that give zero or arbitrary small training error in some machine learning tasks. In particular, we have shown that for the case of two disjoint compact sets, exact fit in a two class classification task is feasible with 4 layers, provided the sets can be separated by a cone like sector. However, there remains a gap between our positive results and limitation implied by the maximum principle. Our numerical experiments show that neural networks learn transformations similar to our theoretical derivations.

## REPRODUCIBILITY STATEMENT

This paper is a contribution to the theory of neural networks, which includes some numerical examples to illustrate our findings. For this purpose, we work only with artificial two-dimensional data and simple network architectures, each of which is described in detail and is therefore easily reproducible. Further, we provide a code implementation in the supplementary material to reproduce the results of our paper. A proof is given for all the theorems, propositions and lemma either in the main paper or in the appendix.

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

## A  APPENDIX

### PROOF PROPOSITION 2

*Proof.* We start with the proof of (1). It is clear that $P_U$ followed by a linear isomorphism $E : U \to \mathbb{R}^m$ with $m = \dim(U) < n_0$ can be implemented by $x \mapsto W_1 x + b_1$ with some $W_1 \in \mathbb{R}^{m \times n_0}$, $b_1 \in \mathbb{R}^m$. As $M$ is compact and hence bounded, one can further arrange $b_1$ in a way that $W_1 x + b_1 \geq 0$ holds (component wise) for every $x \in M$, which implies that $\mathrm{ReLU}(W_1 x + b_1) = W_1 x + b_1$ for all $x \in M$. Thus $g(x) = \mathrm{ReLU}(W_1 x + b_1)$ maps $M$ bijectively to $M_1 := \mathrm{ReLU}(W_1 M + b_1) \subset \mathbb{R}^m$. Now, uniform approximation of a given continuous function $f$ from $M$ to $\mathbb{R}$ amounts to uniform approximation of continuous function from $M_1$ to $\mathbb{R}$. By Tietze's Extension Theorem (Rudin, 2006) we can apply Theorem 1 from Hanin & Sellke (2017) which yields arbitrary accurate uniform approximation of continuous function on rectangular domains in $\mathbb{R}^m$ by network functions in $\mathcal{N}\mathcal{N}_{\mathrm{ReLU}}(m + 1)$.

To show (2), let $W_1 \in \mathbb{R}^{1 \times n_0}$ and $b_1 \in \mathbb{R}$ such that $y_j = W_1 x_j + b_1$, $j = 1, ..., m$, are distinct points $M_1 = \{y_1, ..., y_m\} \subset [0, \infty)$. It is then clear that also $F_0(x) := \mathrm{ReLU}(W_1 x + b_1)$ maps $M$ to $M_1$. Now, for some function $f : M \to \mathbb{R}$ (which is automatically continuous as $M$ is finite) we have to find a network function $F_1 \in \mathcal{N}\mathcal{N}_{\mathrm{ReLU}}(2)$ such that $F_1(y_j) = f(x_j)$ for $j = 1, ..., m$. To this end, we observe that the mapping $y_j \mapsto f(x_j)$ $j = 1, ..., m$ can be interpolated by a max-min string, see Hanin & Sellke (2017, Definition 1) for a definition, so that Hanin & Sellke (2017, Proposition 2) yields the desired $F_1$. Alternatively, one could follow the proof of Hardt & Ma (2017, Theorem 3.2.), observing that on the non-negative $y_j$, $j = 1, ..., m$, the residual $\mathrm{ReLU}$ network function constructed in the latter can be written as a $\mathrm{ReLU}$ network function in the sense of this work with width two in our case. Finally, we have that $F = F_1 \circ F_0 \in \mathcal{N}\mathcal{N}_{\mathrm{ReLU}}(2)$ and $F(x_j) = f(x_j)$ for all $j = 1, ..., m$. □

### PROOF PROPOSITION 4

*Proof.* It is clear that the existence of inner points in $F(M)$ requires that every weight matrix of $F$ is square and of full rank. Indeed, otherwise the range of $M$ is reduced to a manifold of dimension less than or equal to $n_0 - 1$ after a layer that violates this condition. From then on the range remains without inner points. If $L = 1$, then we are done, since in this case $F$ reduces to an isomorphism on $\mathbb{R}^{n_0}$. Otherwise, let $M_j := A_j(M_{j-1})$ for $j = 1, ..., L$ and $M_0 := M$. Hence $F(M) = M_L$. Since $W_L$ is non-singular, $u_L \in M_L$ is an inner point of $M_L$ if and only if $u_L = W_L u_{L-1}$ for some unique point $u_{L-1} \in M_{L-1}^\circ$. This implies that component wise $u_{L-1} > 0$ and further, by the fact that $W_{L-1}$ is non-singular, $u_{L-1} = W_{L-1} u_{L-2} + b_{L-1}$ for some unique point $u_{L-1} \in M_{L-2}^\circ$. The exact same reasoning can now iteratively be applied down to the first layer. This gives a sequence of $u_j \in M_j^\circ$, $j = 1, ..., L$ so that component wise $u_j > 0$ holds for every $j$. Hence, $u_j = W_j u_{j-1} + b_j$ for $j = 1, ..., L$. Thus, for the inverse image $B = F^{-1}(F(M)^\circ) \subset M^\circ$, the mapping $F$ reduces to an bijective linear affine mapping. □

NUMERICAL EXPERIMENT

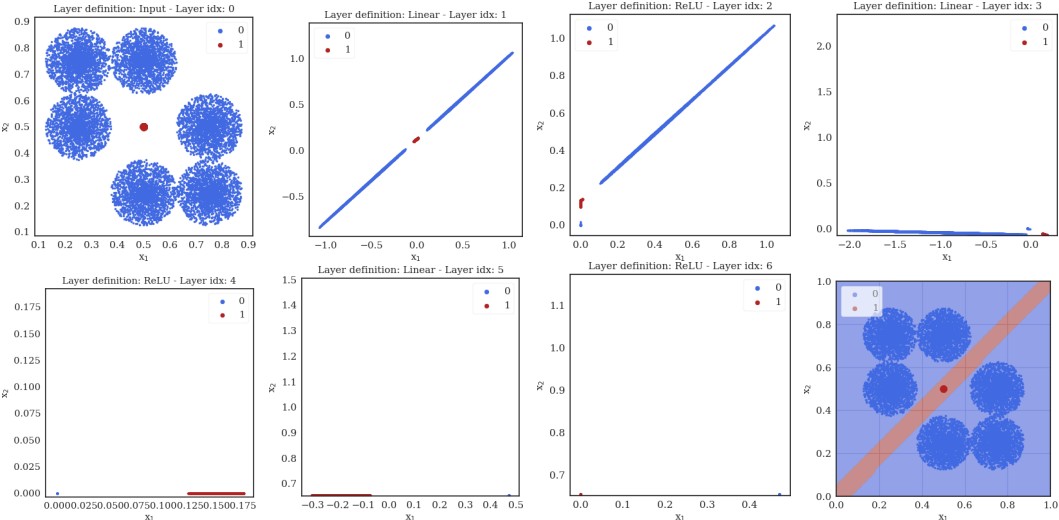

(a) 6 border balls - center ball radius: 0.01 - MSE: 0.0 - UUAC: 0.0

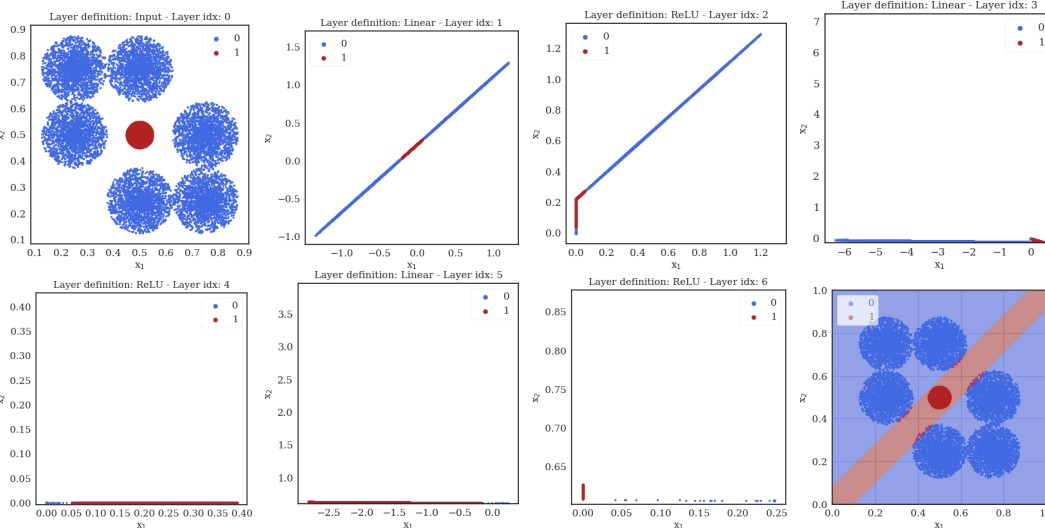

(b) 6 border balls - center ball radius: 0.05 - MSE: 0.00026 - UUAC: 0.99092

Figure 2: Additional results for Fig. 1 of the main paper. Different input datasets (a) and (b) together with the resulting transformations by each activation function and each layer learned by the neural network after 500 epochs. We also report the final decision regions, the mean-square error (MSE) and the universal uniform approximation condition (UUAC). When the conditions of Theorem 2 are fulfilled (a), the learned transformations are similar to the construction in the proof of Theorem 2. When the condition of Theorem is violated (b), the intermediate layers also violate the construction in the proof. The title of each subplot describes the function which is applied to the data of the previous subplot to form the data of the current subplot.

