# OpenReview forum: "Expressiveness of Neural Networks Having Width Equal or Below the Input Dimension"
_ICLR.cc/2022/Conference — ICLR 2022 Submitted_

### Official Review · Reviewer_oqnR · 2021-10-30

**Correctness:** 3
**Technical Novelty And Significance:** 2
**Empirical Novelty And Significance:** Not applicable
**Recommendation:** 6
**Confidence:** 3

**Main Review:**

# Major observations

In Section 4, repeating what I said above, the authors claim through Theorem 2 a result applicable to a "finite collection of disjoint n0-dimensional compact sets" However, except for Corollary 2, which is not explicitly linked to Theorem 2, the entire discussion for the second contribution focuses on only 2 sets. Hence, in Section 6, it is not clear how they claim to be validating the result by using domains consisting of more than 2 disjoint sets.

In Theorem 2, do the authors really mean $K_2 \subset \mathbb{R}^{n_0} \setminus \overline{S}$ instead of $K_2 \subset \mathbb{R}^{n_0} \setminus S$?

# Minor observations

1) Introducion

", we refer to (DeVore et al., 2020) for an overview on recent developments": use semicolon (;)

"‖·‖is the Euclidean norm": do not start a sentence with something other than a word

"j-te coordinate axis": j-th

I can follow your notation, but I strongly encourage you to describe what L is at the begining instead of at the end, at which point you have already used it to define a number of other things.

Additional references:
- On the discussion of expressiveness in terms of depth and width: [1] (figure 5)
- On the ability of memorization in terms of expressiveness: [2] (figure 5, left)

2) Related work

The meaning of $C$ and $L_p$ is never mentioned; neither what exactly each term is in the notation that you start using here. For example, that the first term in the tuple is the domain and the second is the image encompassing the family of functions.

"multiply pairwise compact components": multiple

3) Maximum principle

Remind the reader what "NNσ(n0)" stands for.

"on arbitrary compact sets is impossible:": does not seem to be the right place to use a comma

Briefly repeating the meaning of technical terms can make wonders to accessibility.
In Theorem 1, I would recommend expaning the statement as "takes its maximum value *at the boundary* ∂M".

"to network function of width larger than the input dimensions": confusing; please rewrite

"straight forward": straightforward

"In Theorem 1 show that": Theorem 1 shows that

4) Approximation properties on subsets

"on such kind of sets": that seems to imply whether it applies to each set individually; perhaps something like "on such domains" would be better here

7) Conclusion

"haven shown": have shown

"with cos as activation function": with cosine as activation function

A) Appendix

Proof of Corollary 2: what do you mean by "Proposition 1.2"?

# References cited

[1] https://arxiv.org/abs/1711.02114

[2] https://arxiv.org/abs/1906.00904

**Summary Of The Paper:**

This paper presents three results concerning the universal approximation of functions on compact sets under a width regime for which universal approximation in general is not possible. The authors make a convincing case that this represents an important frontier both for applications and for a better understanding of what neural networks can represent. While I appreciate the first result, I am confused with the second and I do not think the third is relevant.

These results are the following:

1) The maximum principle

This is an interesting observation about the maximum (and minimum) value achieved by neural network with continuous monotonic activation and width bounded by the input layer on a compact domain is always at the boundary of such domain. Trying to break their result made me understand and agree with their claim that this could be understood as a "root cause why universal approximation with functions from NNσ(n0) on arbitrary compact sets is impossible".

2) The result on disjoint domains

In Section 4, the authors claim through Theorem 2 a result applicable to a "finite collection of disjoint n0-dimensional compact sets". Notably, most of their work there focuses on exact representation, which reflects the fact that they are considering functions that are constant on each disjoint set. We can reasonably associate that to classification tasks, but then we cannot assume any probabilistic meaning with something like a softmax layer.

3) The result on harmonic activations

The authors shows that universal approximation can be achieved on a finite and discrete domain with the composition of cosine functions. This result lies entirely on the domain being discrete and finite, which actually breaks any dependence to the width of the input layer. For that reason, I do not think that it is that relevant because it has nothing to do with the main theme of the paper.

**Summary Of The Review:**

In full disclosure, I am curious to know what the other reviewers think of the first contribution, but at this point this is the most positive part of the paper in my view.

However, the organization and discussion of the second contribution really confuses me. I am not comfortable with the current organization of the results related to Theorem 2 and the corresponding experiments in Section 6.

---

> ### Author Response · Authors · 2021-11-11
> **Answer to reviewer**
>
> Thank you very much for your careful review and valuable comments on our work.
>
> One of your main criticism targets the setup in our experimental results in relation to the conditions in Theorem 2. To address this point, we replaced Figure 1 (a) in section 6. Now, the interplay between Theorem 1 and Theorem 2 is depicted more clearly. In case of Figure 1 (a) exact fit is possible, because one set is only nearly enclosed by a second set, but leaves a cone like connection to infinity (as stated in Theorem 2). On the other side it becomes impossible in the case depicted in Figure 1 (b), which corresponds to the result of Theorem 1. However, we would like to note that our first plots were also in line with the conditions stated in Theorem 2:  we do not assume $K_1$, $K_2$ to be connected. We also changed the text that motivates Theorem 2 in a way that it is clear now that two disjoint sets are mainly considered instead of multiple such sets. We also replaced the word “component” by the word “sets” to avoid that the reader implicitly assumes the sets to be connected.
>
> We agree with your statement that Theorem 3 lacks a direct connection with Theorem 1 and Theorem 2 and therefore we removed this result for the sake of a common thread, which is the interplay between Theorem 1 and Theorem 2 (c.f. experimental results). Instead we included some missing material from the appendix.
>
> Regarding your question on whether we mean $K_2$ without the closure of $S$, or $K_2\setminus S $. In fact, both can be used here since, by the compactness of $K_1$, the vectors used to define $S$ can be slightly changed without changing something important for the setup.

---

> > ### Comment · Reviewer_oqnR · 2021-11-19
> > **Following up**
> >
> > I appreciate the careful review of the paper by the authors, and in particular the exclusion of the third result given that it was disconnected from the rest and it was not as relevant for the work. I also appreciate your follow up on my question.
> >
> > I am a bit more confident on the relevance of this work and I am updating my score accordingly. With that said, I am relying on other reviewers validating the correctness of Theorem 2, which is not exactly in my comfort zone.

---

### Official Review · Reviewer_PCRn · 2021-10-30

**Correctness:** 4
**Technical Novelty And Significance:** 3
**Empirical Novelty And Significance:** 3
**Recommendation:** 5
**Confidence:** 4

**Details Of Ethics Concerns:**

-

**Main Review:**

The reviewer finds the topic studied in the paper very interesting. Expressivity bounds have been studied to understand benefits of depth. However, the case of width is also interesting and appears to be less studied in the literature. The basic fact here is that universal approximation is impossible if the width of a network is severely restricted to be less than the input dimension $n_0$. Given this, I find the question studied in this work well-motivated and natural.

The only concern with the present work is the novelty. Even though it is technical, its main message is not too strong. The reviewer does not follow how their "results give theoretical insights on the kind of feature extraction earlier layers need to implement in order to allow the later layers to solve a givem machine learning task", as they state in Section 2. Moreover, the reviewer thinks that the paper has an overall nice collection of (relatively) small results, but none of them is particularly strong. The maximum principle derived is perhaps the most interesting; however, it seems incremental given the Beise et al. (2021) studying the decision regions of narrow deep NNs and Johnson (2018). The other two contributions about width-restricted ReLU nets of depth 4 and cosine nets of depth 3 are nice but not in par with an ICLR paper.



**Summary Of The Paper:**

The paper studies the question of expressivity of neural networks that have restricted width. Specifically, the main question is "what happens when the maximum (among all layers) width is less or equal to the input dimension?"

From prior works, it is known that such width-restricted networks cannot possibly express all continuous functions on arbitrary compact sets. So the next natural question is what kind of functions/subsets M of R^n are amenable to approximation via such networks?

The main contribution is the derivation of certain topological conditions on compact sets M in R^{input dimension} that allow or exclude universal approximation by such networks.

First, the authors derive a maximum principle stated as Theorem 1: If M is a compact subset of R^{input dimension}, then a neural network of width at most the input dimension, equipped with any continuous monotonic activation function  will necessarily attain its maximum value on the boundary of M. As a consequence, the authors show how previous lower bounds on width (w\ge n_0 +1) are tight by providing several tight examples.

Second, the authors investigate the case where the domain consists of two disjoint compact sets and the goal is to approximate functions that take constant values on each of these sets. They give a sufficient condition under which shallow ReLU nets of width $w=n_0$ (the input dimension) and depth 4 can expreess them.

Third, they also consider some expressivity properties of simple networks with cosine activations that have width only 1 and depth 3.



**Summary Of The Review:**

Nice motivation about the question, but the results presented are relatively weak. There is not a "main" message of the paper that is in par with ICLR standards.

---

> ### Author Response · Authors · 2021-11-11
> **Answer to reviewer**
>
> Thank you very much for your careful review and your helpful remarks.
>
> If we understand you correctly, your main concern is that you question the novelty of our result in Theorem 1. It is true that our result excludes universal approximation under the investigated conditions and that this is already shown in the works you mentioned. However, we believe that our result makes important steps beyond these insights:
> 1.	By providing an easy criterion for conditions that exclude universal approximation, the result gives hints for which configurations one can still hope to find (some kind) of universal approximation properties (c.f. Theorem 2).
> 2.	Theorem 1 constitutes a mathematical tool for future works in the bounded width setting.
>
> We therefore think that our work makes a reasonable step and deepens the understanding on results given in Beise et al (2021) and Johnson (2018).
>
> To structure and highlight our main message more clearly, we removed Theorem 3 which, admittingly, lacks a tight connection to Theorem 1 and Theorem 2, and instead we included missing material from the appendix. We also replaced Figure 1 (a) in section 6. Now, the interplay between Theorem 1 and Theorem 2 is depicted more clearly. In case of Figure 1 (a) exact fit is possible as stated in Theorem 2, whereas it becomes impossible in the case depicted in Figure 1 (b) as it is theoretically shown in Theorem 1. Notice that the subsets do not need to be connected and that the experiment investigates two subsets (class 0 and class1), as investigated similarly in Theorem 2.
>
>
> You also say that you cannot follow our reasoning that says “results give theoretical insights on the kind of feature extraction earlier layers need to implement in order to allow the later layers to solve a givem machine learning task”.  What we mean is basically this: In a standard CNN, the first convolutional layers extract features that are, as a whole, located in a comparable high dimensional space, before they are processed by some dense layers. In many usual cases, those dense layers map their input to lower dimensional spaces so that, considering only the processing of these dense layers, we are in the setting investigated in our work.

---

### Official Review · Reviewer_qHTG · 2021-11-01

**Correctness:** 4
**Technical Novelty And Significance:** 3
**Empirical Novelty And Significance:** 3
**Recommendation:** 6
**Confidence:** 3

**Main Review:**

The manuscript has three main contributions: (1) A maximum principle when the activation function is continuous and monotonic (such as ReLU), showing that the neural networks must be at least n_0 + 1 wide, (2) An interesting condition on two compact sets (one of them being contained in a cone-like sector that doesn’t intersect with the other) that is sufficient to allow exact fits of piecewise constant functions, and (3) A width 1 and depth 3 neural network with cosine activation is a universal approximator for a finite sample set.

Strengths:
 - The paper is well written and easy to follow. The discussion regarding prior work is quite clear and extensive.
 - The authors are investigating an unusual question by restricting the types of compact sets to prove universal approximation theorems. Therefore, I think the manuscript has significant merit in the way that it is investigating universal approximation theorems.
-  I agree with the authors that universal approximation on certain subsets is often what is needed in practice.

Weakness:
 - Some of the theoretical results only apply to monotonic activation functions, which covers many important activation functions, but certainly not as general as related recent papers on deep narrow neural networks.

**Summary Of The Paper:**

In the mathematical theory of neural networks, universal approximation theorems typically establish the density of a class of neural networks within a certain function space. In recent years, it has been shown that a width larger than the input dimension is needed to allow universal approximation theorems for continuous function spaces defined on arbitrary compact sets.  In this paper, the authors investigate what kind of subsets of R^n allow for a universal approximation theory with neural networks that have a width <= input dimension.



**Summary Of The Review:**

In my opinion, the article is asking an interesting question by restricting to certain compact sets. I believe that the manuscript is making a reasonable theoretical contribution to the analysis of deep narrow neural networks.

---

> ### Author Response · Authors · 2021-11-11
> **Answer to reviewer**
>
> Thank you very much for your careful review. We appreciate your positive appraisal of our results.
> Regarding your comment on a weakness of our work, we agree that non-monotonic activation functions deserve to be investigated in the context of our presented results. However, is it clear that the maximum principle in Theorem 1 is false in the general case of non-monotonic activation functions, and the proof of Theorem 2 relies entirely on the properties of the ReLU activation function. A generalization of the latter to non-monotonic activation functions is out of scope within this work.

---

> ### Comment · Reviewer_qHTG · 2021-11-27
> **Responding to the authors**
>
> I wish to thank the authors for their response to my review. My opinion of the paper remains unchanged.

---

### Official Review · Reviewer_99iL · 2021-11-03

**Correctness:** 3
**Technical Novelty And Significance:** 3
**Empirical Novelty And Significance:** Not applicable
**Recommendation:** 6
**Confidence:** 4

**Main Review:**

Strengths :

The narrow network regime is not as well studied and is a useful pursuit to obtain a complete characterization of NNs and this paper definitely takes some steps in that direction.

Section 3, which describes the maximum principle is interesting as it talks about the existence of compact subsets that cannot have an universal approximation guarantee when the activation functions are monotonic and continuous, even if you consider classes of NNs with arbitrary depth!

Section 4, talks about subsets that can guarantee exact interpolation and universal approximation guarantees. They specifically consider subsets that are disjoint union of a finite number of compact sets and functions that are constant in each disjoint piece. This can be considered a large class of functions that are used for clustering/classification applications. They provide some sufficient conditions that these sets need to satisfy in order for exact interpolation by a constant depth ReLU network (for two compact sets and for more than two, there exists some depth (input dependent) that can give exact interpolation.

Finally, they show that an arbitrary finite sample set can be exactly interpolated by a constant width and depth network that has a cosine activation function.

Weaknesses:

Theoretical:

My main issues is about the approximable subsets. The authors consider studying functions that are constant over disjoint compact sets, akin to clustering/classification type applications. They present results that target exact interpolation in Theorem 2 and Corollary 2. Now can the assumptions on mutual locations be removed if targeting universal approximation and allowing the depth to be arbitrarily large ? Or is it that in this case exact interpolation is as hard as universal approximation?

In addition, I was wondering about the applicability of theorems in Sections 3 and 4 with a cosine activation function which is not a monotonic function, but does well on an arbitrary finite set (Theorem 3)? It would be nice to have a discussion to consider the applicability of non-monotonic activations as well.

As a general comment, when using NN($n_0$) to represent the class of functions, if possible it would be useful to mention the depth (which may be input dependent) to understand the dependence on the depth which is able to approximate/interpolate the target functions.


Presentation and minor issues:

Another issue is about the presentation of results. I think the paper would be much better with illustrations of sets that help reader visualize the geometry, as these sets are interesting examples/counter-examples.

Minor suggestions:
1) In the Introduction ..."An important results states that a width larger..", is stated without mentioning where the result appeared (which is mentioned in the next section though).
2) In  page 4, when the authors mean $\sigma$ is partially constant, do they mean piecewise monotonic? It would be nice to have the statements of all the auxiliary theorems and lemmas that are not common to be in the appendix for easy reference.

**Summary Of The Paper:**

The paper focuses on approximation of NNs in the narrow regime, i.e, when the width of the network less than the input dimension ($n_0$) and asks the question:
What compact subsets M of $R^{n_0}$ still allow universal approximation or exact interpolation for all functions that map from M to R.?

It first shows the existence of compact subsets that do not allow for such universal approximation guarantee.

Then the paper presents sufficient conditions on certain compact subsets so that some universal approximation/exact interpolation guarantee can be attained and these results are targeted for clustering/classification applications. The functions here are piecewise-constant on disjoint compact sets.

The assumptions on the activation functions are general which includes most of the commonly used ones in practice as well.

**Summary Of The Review:**

I think the problem is definitely interesting, to understand the limitations of narrow networks and of course the extent of positive results that can be extracted from it. I think this work takes some steps in that direction, but definitely requires a bit more investigation (as specified above) and also the presentation could be made better with some illustrations/visualization of the sets in the main results, which may help communicate the ideas in a better way.

---

> ### Author Response · Authors · 2021-11-11
> **Answer to reviewer**
>
> In your first question, wherein you ask whether the conditions on the mutual location of the compact sets in Theorem 2 can be removed when arbitrary depth is allowed, you resume one of our main open question. It is clear that the maximum principle in Theorem 1 provides topological conditions where the answer is no. However, we believe that it is hard to give a general answer that characterizes the conditions that lie between those in Theorem 2 and Theorem 1 that admit exact fit on disjoint compact sets.
>
> In your second comment you ask for a discussion on non-monotonic activation functions regarding our first results (Theorem 1 and Theorem 2). We agree, that this is an interesting open question. Regarding the maximum principle in Theorem 1, we think it is clear that it does not hold anymore without the monotonicity condition, and we will mention this in our next version. Regarding Theorem 2, we cannot conjecture how an analog result for the case of non-monotonic activation functions would look like as our proof entirely relies on the properties of the ReLU activation function. Hence, we think that this is out of scope for this work.

---

> > ### Comment · Reviewer_99iL · 2021-11-29
> > **Acknowledgement of reading the author's response**
> >
> > Thank you for the responses to the comments. Some of my questions are clarified. And after reading the other reviews and responses, I believe that the maximum principle is the most interesting result (but similar results appear in Beise et al 2021). It would have been good to see more results for non-monotonic functions as the message in the later parts of the paper seem to show that non-monotonic functions overcome many limitations shown in earlier sections by monotonic activation functions. My score remains unchanged.

---

### Decision · Program_Chairs · 2022-01-20

**Decision:**

Reject

**Comment:**

*Summary:* Study expressive power of narrow networks.

*Strengths:*
- Study the narrow setting, which is not as well studied as the wide setting.
- Some reviewers found the paper well written.

*Weaknesses:*
- Restricted class of targets and activations.
- Similar results have appeared in previous works.

*Discussion:*

99iL asked about the possibility to remove certain assumptions and the extension to other activations. Authors answer negatively to both. 99iL acknowledges the response and concludes the so-called maximum principle is the most interesting result, but also points out that similar results appear in previous work and that it would have been good to see some extensions. qHTG indicates that the paper is well written and has interesting contributions but that some of the theoretical results only apply in settings that are more restrictive than in other recent related works. Authors agree that generalizations deserve to be investigated in the context of the presented results, but point out that their principle does not apply in that case, and hence that such generalizations are out of scope. Although qHTG identifies several good aspects in this work, they maintain the overall assessment of just marginally above the threshold. PCRn finds the work very interesting but is concerned about the novelty and points out that although the work is technical, the main message is not very strong and that the extraction of insights to solve tasks is not as clear. PCRn concludes that the paper presents various relatively weak results but not a sufficiently significant message. Authors remark that some of their results constitute a mathematical tool for future works.

*Conclusion:*

One reviewer rated this work marginally below the acceptance threshold and three other marginally above. Considering the reviews and the discussion, I conclude that this paper obtains a few interesting results but leaves much for future work. Further development of the current results would make the article significantly stronger. In view of the very high quality of other submissions to the conference, I find that this article tightly misses the bar for acceptance. Therefore I recommend to reject this article. I encourage the authors to revise and resubmit.